# Palestinian Refugee Youth in Jordan: Parental Practices, Neighborhood Cohesion and Assistance, and Adolescent Wellbeing

**DOI:** 10.3390/ijerph18073649

**Published:** 2021-03-31

**Authors:** Ikhlas Ahmad, Judith Smetana

**Affiliations:** 1Human Development Program, Counseling and Educational Psychology Department, University of Jordan, P.O. Box 13343, Amman 11942, Jordan; 2Department of Psychology, Meliora Hall, RC 270266, University of Rochester, Rochester, NY 14627, USA; smetana@psych.rochester.edu

**Keywords:** post-war, parental practices, adolescent wellbeing

## Abstract

In this study, a total of 335 Palestinian refugees (*M* = 15.5 years, *SD* = 1.05, 49% males), recruited from four United Nations Relief and Work Agency (UNRWA) schools at the Al-Baqa’a and Jabal Al-Hussein refugee camps in Jordan, rated their neighborhood physical environment and neighborhood support and cohesion, separately rated their mothers’ and fathers’ parenting on several dimensions, and reported on their adjustment to these circumstances (internalizing symptoms, self-concept clarity, and norm breaking). Living in more dangerous physical environments was associated with higher levels of refugee youths’ internalizing symptoms and norm breaking, but effects were not significant when parenting was considered. Our study showed that higher levels of psychological control–disrespect (significantly for fathers and marginally for mothers) and marginally, higher levels of maternal harsh punishment were associated with more teen internalizing symptoms. In addition, fathers’ greater psychological control and lower levels of support had a marginally significant effect on teens’ greater norm breaking. For behavioral control, only mothers’ greater behavioral control was associated with refugee youths’ greater self-concept clarity but not with paternal behavioral control. Thus, fathers’ psychological control and mothers’ behavioral control had the biggest association with adolescent outcomes.

## 1. Palestinian Refugee Youth in Jordan: Post-War Parental Practices and Adolescent Wellbeing

The major increase in the number of refugee families in the Middle East over the past decade has led to increased interest in Arab refugee youth’s welfare and mental health. Many of these families have not been directly displaced; instead, they have resided in refugee camps for several generations. This is the case for many Palestinian families, and youth in these families are experiencing various forms of internalizing and norm-breaking problems [1]. Residents of refugee camps typically experience poverty, poor nutrition, and overcrowded neighborhoods [2], and all of these may place many strains on families and influence family dynamics and parenting practices. 

The present study investigated associations among neighborhood physical and psychological conditions, parental disciplinary practices, and adolescent adjustment in a sample of Palestinian refugee youth living in Jordan. Jordan has received many refugees, particularly Palestinians, and many have lost hope or even the belief in goodwill from other Arab peoples. Palestinian refugees feel demoralized from neglect and victimization [3], and this contributes to their loss of human dignity. Among such circumstances, punitive parental practices and the harsh conditions of life in refugee camps can intensify the already stressful lives of refugee families [4] and contribute to youths’ adjustment problems.

### 1.1. Parental Practices in Arab Culture

Past research has shown that parenting varies across Arab societies [5], with parenting in more traditional countries emphasizing more control than in more modern countries such as Lebanon and Jordan [6,7,8]. Arab families have generally been described as authoritarian, although some have asserted that parenting styles in the Arab world are not as distinct from that in the West [8,9]. Arab fathers and mothers generally promote parental authority and control over their children’s behaviors [10]. R = Recent research has shown that Arab refugee parents from several different societies are seen by teenagers as having legitimate authority to regulate a variety of their behaviors (friendships, moral issues, conventional norms, and health, safety, and risky behavior), but not personal issues [11].

Parents in Arab cultures tend to be psychologically controlling [12], and the negative effects of parental psychological control on children’s and adolescents’ development, including greater internalizing and externalizing problems, has been consistently demonstrated in countries around the world [13]. Psychologically controlling parents attempt to obtain obedience via manipulating their children’s thoughts and feelings [14]. These parents can intrude on the child’s psychological and emotional development by invalidating feelings, withdrawing love, and inducing guilt [14]. More recently, Barber and his colleagues [15] have identified parents’ disrespect of the child as the most harmful aspect of psychologically controlling parenting. 

In addition, harsh parenting also tends to be prevalent in Arab families. A study conducted in Israel with a large sample of Arab adolescents indicated that 52% had experienced harsh parenting, including being beaten with a stick, club, or other harmful objects [16]. Harsh parenting may involve yelling, negative emotional expression, name calling, overt expression of anger, physical threats and aggression, or physical punishment [17]. Not surprisingly, harsh parenting leads to a range of negative outcomes, such as externalizing problems for children, including aggressive and antisocial behaviors, as well as poor mental health [18,19]. Several studies found a greater use of parental harsh discipline toward boys than toward girls [20,21]. Nonetheless, research on Arab cultures and harsh discipline during adolescence confirmed that it is often more focused on girls than on boys, especially on disciplining girls who commit a violation of their family honor [22].

Although psychological control and harsh parenting are common in Arab families, past research has also shown that Palestinian adolescents view their parents as employing behavioral control [23]. Behavioral control has been distinguished from psychological control and focuses on managing adolescents’ behaviors through regulating their actions, such as setting rules, supervising activities, and enforcing household rules. Therefore, at least at moderate levels, behavioral control can be seen as a positive parenting practice that facilitates better child adjustment, but at high levels, it may be interpreted as intrusive and psychologically controlling [24]. For instance, the absence of adequate behavioral control increases externalizing problems for adolescents, while the absence of psychological control may lead to better social and emotional adjustment [25].

### 1.2. Palestinian Refugee Families in Context

Typically, the family is recognized as the center of economic and social interactions in Arab culture, including Palestinian refugee families [26]. However, refugee status and low income may undermine parents’ position among family members. Palestinian refugees are concentrated at the lower socioeconomic levels and often live in poverty. Arab parents encourage the interdependence, rather than the independence, of individual family members; in such a setting, social class is considered a fundamental key for decision-making processes. Social class is found to have an impact on Arab family structure—regarding authority and family hierarchy—particularly among lower-class refugees in Jordan [21]. Refugee families often lack adequate health care and education, and in addition, Palestinian youth have few job opportunities.

Refugee youth and parents are also affected by the condition of their physical environment. Youth who reside in refugee camps typically live in harsh physical environments and unsafe conditions [21,27]. Prior research on children affected by such physical conditions showed an increased risk of mental health problems ranging from adjustment difficulties to depression and anxiety disorders. Most importantly, many refugee families work long hours in order to ensure that their children receive an education. Therefore, the social environment of their neighborhoods, including neighborhood cohesion and assistance, may be an important factor that is associated with youth adjustment.

Relatively few studies have examined how parenting and living conditions are associated with refugee adolescents’ wellbeing. However, a great deal of previous research has shown that parents who live in dangerous neighborhoods in low- and middle-income countries around the world tend to employ harsh parenting, which in turn is associated with poor adjustment [28,29]. Thus, it is likely that growing up in Arab refugee families, where parents employ harsh punishment and psychologically controlling parenting, and where youth live in unsafe physical environments, may have negative consequences for youths’ development.

### 1.3. Palestinian Mothers’ Versus Fathers’ Parenting

Most past studies on Arab parenting do not distinguish between fathers’ and mothers’ parenting [30]. However, due to culture and religion, Arab fathers and mothers may treat their children differently. Patriarchal ideology is rooted in Arab societies [31]. The role of Middle Eastern fathers is to ensure that the family’s physical and economic needs are met [32], whereas Middle Eastern mothers typically have more responsibility for child rearing, discipline, education, and household duties. Therefore, adolescent misbehavior is sometimes blamed on the mother, and the adolescent’s behavior is seen as a reflection of the family’s reputation.

Still, in Arab families, the father is considered to be the head of the family, while the mother and other young family members are expected to obey the father’s authority. In a study using a Saudi sample, fathers tended to be more controlling than mothers, and the differences between fathers and mothers’ control were obvious in our study; indeed, the father’s control may have a negative impact on adolescents’ adjustment, including low self-worth. However, there were no such associations for mothers’ parenting styles, including support [33]. Moreover, in both traditional and modern Arab societies, parents tend to be more authoritative and less authoritarian toward girls than toward boys [34]. Nevertheless, when fathers practice authoritative parenting, this often has a positive impact on the family, as shown in a study in the West Bank, Gaza [35], in terms of communication and academic achievement. Another study [36] using a Palestinian sample from the West Bank showed that less-educated parents were more overprotective or authoritarian regardless of parent gender.

### 1.4. The Present Study

The present study contributes to the literature by examining Arab Palestinian refugees’ perceptions of mothers’ versus fathers’ parenting, including parental support, psychological control–disrespect, behavioral control, and harsh parenting, and their associations with adolescents’ adjustment—internalizing symptoms, self-concept clarity, and norm breaking. We examined the relative importance of fathers’ versus mothers’ parenting in contributing to refugee adolescents’ adjustment along these dimensions. We also examined whether neighborhood safety and physical environment, as well as neighborhood cohesion and assistance, were associated with adolescent adjustment.

In Arab society, religion also plays an important role in raising adolescents, where it emphasizes that adolescents obey their father and mothers equally. Arab parents tend to be supportive of protecting their adolescents’ wellbeing. Consistent with past research, we expected that greater parental support and behavioral control would be related to higher levels of psychosocial functioning (e.g., self-clarity) and lower levels of maladaptive behaviors (e.g., psychological internalizing symptoms and problem behaviors [37,38,39]). Having support from parents (mother or father) is expected to have a positive impact on adolescents’ wellbeing, regardless of the gender of parent.

Consistent with past research, we expected both harsh parenting and parental psychological control would be linked with more problematic adjustment, including more internalizing distress and problem behavior [13,15]. More specifically, because of the high status that fathers have in Arab families in comparison with mothers, we predicted that fathers’ psychological control and harsh punishment would be more strongly associated with negative outcomes, including more internalizing problems, less self-clarity, and more norm breaking, than would mothers’ use of these parenting practices.

We also examined the role of neighborhood support (neighbors’ willingness to respond to others) on adolescents’ adjustment. We hypothesized that while living in poverty and dangerous neighborhoods, parents may control their children to keep them out of trouble. In sum, we predicted that, based on cultural roles, fathers and mothers would have a different impact on their adolescents’ outcomes after taking neighborhood variables into consideration.

## 2. Methods

### 2.1. Participants and Procedure

The sample for this study includes 335 Palestinian refugees (165 boys and 170 girls) ranging in age from 12 to 18 years old (*M* = 15.5 years, *SD* = 1.05). They were recruited from four United Nations Relief and Work Agency (UNRWA) schools located in two refugee camps: Al-Baqa’a and Jabal al-Hussein refugee camps in Middle and North Amman (the capital city of Jordan), respectively. The Al-Baqa’a camp is one the three major Palestinian refugee camps in the central Amman region, located at Al-balqa’a area of the country of Jordan; situated in the Asalt area, Jabal al-Hussein is located in the northwest of Amman. The camps face overcrowding, and according to the Fafo Foundation Report (2013), 32% who are living in Al-Baqa’a camp and almost 28% in the Jabal Al-Hussein camp are living below the national poverty line. Based on UNRWA (2016), 46% of Palestinian refugees in Baqa’a and 69% of refugees in Jabel Hussein do not have health insurance. The Baqa’a camp is ranked second out of ten camps in unemployment rates, where17% of refugees living in the camp are unemployed. Upgrading camp infrastructure and shelter repair and rehabilitation are also major challenges. Overall, poverty, high unemployment, and poor infrastructure are major challenges facing these camp residents (UNRWA, 2016). Nearly all of the adolescents in our sample (more than 90%) reported on both mothers’ and fathers’ parenting. As regards educational level, 35% of fathers and 29% of mothers had graduated from high school. There was an average number of 4.52 children in the family. All participants were from poor backgrounds. Mothers gave permission for their adolescents to participate, and adolescents filled out questionnaires at school.

### 2.2. Measures

Following the guidelines of the International Test Commission [40], all questionnaires used in this study were translated from English to Arabic by two college bachelor-level students whose college major was in translation. The back-translation was done by two independent American scholars familiar with Arabic. Comparison of the original questionnaires with the back-translation showed strong convergence. Approval from the University of Jordan Human Subject’s Committee was acquired in order to collect data for the study. The researchers also obtained approval for research on human subjects from the UNRWA, also known in Jordan as the Office of Field in Jordan (OFJ).

Parental Support. Participants rated mothers’ and fathers’ warm and supportive interactions on eight items taken from Barber et al. (2005); e.g., “My mother/father is a person who gives me a lot of care and attention.” Items were rated on a 5-point Likert scale ranging from 1 (strongly disagree) to 5 (strongly agree). Cronbach’s alpha for reports of fathers was 0.85 and for mothers was 0.89.

Parental Psychological Control–Disrespect. Participants rated mothers’ and fathers’ use of psychological control–disrespect using Barber et al.’s (2012) eight-item scale; e.g., “My mother/father ridicules me or puts me down (e.g., saying I am stupid, useless, etc.”). Items were rated on a 5-point Likert scale ranging from 1 (strongly disagree) to 5 (strongly agree). Cronbach’s alpha for ratings of fathers was 0.73 and for ratings of mothers was 0.72.

Behavioral control. Mothers’ and fathers’ behavioral control was assessed using Kerr and Stattin’s (2000) five-item measure. Items (e.g., “My mother/father always requires that I tell him/her where I have been at night, who I was with, and what we did together”) were rated on a 5-point Likert scale ranging from 1 (strongly disagree) to 5 (strongly agree). Cronbach’s alpha for ratings of fathers and mothers was 0.86 and 0.85, respectively.

Harsh Parenting. Adolescents rated mothers’ and fathers’ use of harsh punishment using [41] a four-item measure, which refers to instances of hitting, slapping, shoving, and hitting the child with an object (e.g., “When you do something wrong, how often does your mom/dad spank or slap you?”). Three items were used; responses were rated on a 5-point scale ranging from 1 (Never like my parent) to 5 (Exactly like my parent). Cronbach’s alpha for ratings of fathers was 0.70 and for mothers was 0.71. 

Self-Concept Clarity. Adolescents rated the extent to which they perceive themselves clearly and internally with a stable opinion using [42] an 11-item scale (e.g., “In general, I have a clear sense of who I am and what I am”). Items were rated on a scale ranging from 1 (strongly disagree) to 5 (strongly agree). Alpha was 0.78.

Internalizing Symptoms. Adolescents completed an 18-item shortened version of the Brief Symptom Inventory (BSI; [43]), which assesses symptoms of negative effect, withdrawal, and suicidal ideation. Participants rated how much they were distressed by different symptoms over the past seven days on a 4-point scale ranging from 1 (Not at all) to 4 (Extremely). The mean for all items was used here. Cronbach α was 0.91. 

Neighborhood Physical Environment. Nine items [44] were used to assess the condition of the neighborhood physical environment (e.g., “There are vacant/burned buildings”). Items were rated on a 4-point scale ranging from 1 (strongly agree) to 4 (strongly disagree), with higher scores indicating a poorer environment. Cronbach’s alpha was 0.79.

*Neighborhood Cohesion and Assistance*. We also used 10 items from [45] assessing social cohesion and assistance. There were 6 items assessing neighborhood cohesion (“People in my neighborhood generally get along with each other, look out for each other, can be trusted”), and 4 items assessing assistance (“In my neighborhood, neighbors could be counted on to ‘do something’ if a child or teen was threatening someone with a weapon”). Cronbach’s alpha was 0.85 for cohesion and 0.71 for support. These two scales were moderately correlated, r(328) = 0.44, *p* < 0.001, and were combined *t* form a single neighborhood cohesion and support scale.

Norm breaking. Norm breaking was assessed using Kerr and Stattin’s (2000) 7-item measure (e.g., “Have you taken money from home?” and “Have you been a part of a physical flight?) ranging from 1 (Strongly disagree) to 6 (Strongly agree). Cronbach alpha was 0.78.

## 3. Results

### Descriptive Statistics and Correlations

Means, SDs, and correlations among study variables are shown in Table 1. Adolescent self-clarity, internalizing symptoms, and norm breaking were positively associated with mothers’ behavioral, and psychological control, maternal support, and mothers’ harsh parenting. Maternal support was positively related to adolescent self-clarity and negatively related to adolescent depression and norm breaking. Maternal psychological control and harsh parenting were positively related, and behavioral control was negatively related to adolescent internalizing symptoms and norm breaking, and behavioral control was also positively related to self-clarity. Greater self-clarity and lower levels of internalizing symptoms and norm breaking were associated with paternal support and behavioral control. Paternal psychological control was associated with adolescent internalizing symptoms and norm breaking.

Because we were interested in the effects of different parenting dimensions on our three indices of adjustment, we elected not to use a latent variable approach and instead include the individual parenting dimensions in hierarchical regression analyses conducted separately for the three adjustment variables. Moreover, as research on parenting typically focuses either on maternal parenting or, more generally, on global measures of parents, we first entered adolescents’ ratings of mothers’ parenting into the models and then added adolescents’ ratings of fathers in a separate step to see if the latter contributed to the different adjustment variables over and above the effects of maternal parenting. 

More specifically, adolescent age and gender and parents’ education (i.e., mean levels of mothers’ and fathers’ education) were included in the first step. Because neighborhood conditions are more distal to adjustment than parenting, we entered neighborhood physical environment and neighborhood cohesion/support in the next step. In the third step, we added ratings of mothers’ parenting (support, harsh punishment, and psychological and behavioral control). In the fourth, final step, we added adolescents’ ratings of fathers’ parenting. Results for the final step are shown in Table 2. We describe the results below separately by outcome variable.

Internalizing Symptoms. When neighborhood variables were added at Step 2, neighborhood physical condition (but not neighborhood cohesion/support) was significantly associated with greater internalizing symptoms (*b* = 0.14, *p* < 0.05). However, this effect became nonsignificant when ratings of mothers’ parenting were added to the regression equation. When only mothers’ parenting was considered (at Step 3), both greater harsh punishment and higher levels of psychological control–disrespect were significantly associated with adolescent internalizing symptoms (*b*’s = 0.18, 0.17, *p* < 0.01), and behavioral control was marginally but negatively associated with internalizing (*b* = −0.11, *p* < 0.10). However, when fathers’ parenting was added to the equation, the effect of mothers’ harsh punishment became marginal (*b* = 0.12, *p* = 0.08), mothers’ psychological control became nonsignificant, and adolescents’ ratings of fathers’ greater psychological control–disrespect (but none of the other father parenting variables) were significantly associated with more internalizing symptoms (*b* = 0.22, *p* < 0.001). As expected, girls scored higher on internalizing symptoms than boys.

*Self-Concept Clarity*. Age was negatively associated with self-concept clarity at the first step (*b* = −0.15, *p* < 0.01), but this effect became nonsignificant as other variables were added to the regression equation. Greater maternal behavioral control contributed to greater self-concept clarity, both when only mothers’ parenting was considered and when adolescents’ ratings of fathers’ parenting was considered. Higher levels of fathers’ psychological control–disrespect (*b* = 0.14, *p* < 0.06) were marginally associated with greater adolescent self-clarity after controlling for mothers’ parenting. There were no significant effects for demographic background or neighborhood.

*Norm breaking*. As expected, gender (being male) was significantly associated with greater adolescent norm breaking. Neighborhood physical condition had a significant effect when examined at Step 2 (*b* = 0.13, *p* < 0.05), with poorer neighborhood conditions associated with greater teen norm breaking, but the effect became nonsignificant when parenting variables were added to the regression equation. When only mothers’ parenting was considered, lower levels of mothers’ behavioral control and higher levels of psychological control (*b* = −0.15, 0.13, *p* < 0.05), and marginally, greater harsh punishment (*b* = 0.12, *p* < 0.06) were associated with greater adolescent norm breaking. However, these effects became nonsignificant when fathers’ parenting was included in the model. In the final model, lower levels of fathers’ support and higher levels of fathers’ psychological control (*b* = 0.12, *p* < 0.09) were marginally associated with greater norm breaking (*bs* = 0.12, *p* < 0.08).

## 4. Discussion

The present study extends previous research on associations between Arab parents’ parenting and adjustment by studying Palestinian adolescents living in Jordanian refugee camps and the unique effects of mothers’ versus fathers’ parenting, along with the effects of the broader context (e.g., neighborhood physical environment and support). We examined associations between four well-studied dimensions of parenting—parental support, harsh punishment, and behavioral and psychological control, measured separately for fathers and mothers on Palestinian refugee adolescents’ adjustment, as those four dimensions are the most well-known parental practices in Arab society [1].

Because mothers are usually involved in more of the day-to-day details of parenting than fathers, much of the research on parenting has focused on adolescents’ reports of mothers’ parenting (or has not distinguished the gender of the parent). Therefore, in the current study, we examine whether Palestinian refugee youths’ reports of fathers’ parenting were associated with teen adjustment over and above the effects of mothers’ parenting. Similarly to what has been found in past research cross-culturally, greater behavioral control was related to less norm breaking [46]. Previous studies showed that behavioral control is associated with better adjustment through developing adolescents’ self-regulatory strategies, and greater parental behavioral control had positive effects for the Palestinian refugee youth studied here. However, in addition, we found that the effects of behavioral control differed for mothers and fathers. For mothers, they were also related to greater self-clarity, whereas for fathers, they were associated with lower levels of internalizing symptoms. Parental control and support lead adolescent to cooperate with their parents by providing partial or whole information or even voluntary disclosure after feeling that their needs were met. From a self-determination theory lens, relatedness—feeling that one is connected with parents and matter to others—is also one basic psychological need and nutriment [47].

Adolescents could be more vulnerable to adjustment difficulties when they feel emotionally disconnected from their parents; this may lead to greater norm breaking and internalizing symptoms, and reduced self-clarity. Adolescents who interpret their mothers’ and fathers’ control as negative tend to be less accepting of their control. When adolescents interpret their fathers’ control as supportive, they experience feelings of connectedness to their parents. However, when adolescents understand their parental control in a supportive way from their fathers, they experience feelings of being controlled and connected to their parents [48]. Further, one study [49] found that youths whose parents were less accepting and more rejecting felt that they mattered less to their parents. Although we did not test this interpretive process here, it may explain the link between their father’s behavioral control and positive adolescents’ adjustment.

Similarly to what has been shown in past research, we found that mothers’ and fathers’ harsh parenting is associated with greater internalizing symptoms. Adolescents who learn from their parents to use physical coercion to control relationships are more likely to use those behaviors in social interaction with others or within their families [50].

Generally, psychological control adversely affects one’s self-image through inhibiting youths’ psychological autonomy and potentially harming the core of self-identity. Here, we found that mothers’ and fathers’ psychological control had different effects on adolescents’ outcomes. Fathers’ psychological control had a broader impact in comparison with that of mothers. That is, fathers’ psychological control was strongly associated with poorer adjustment on all the adjustment outcomes studied here, whereas mothers’ psychological control was associated only with greater internalizing symptoms. This may reflect standard gender roles in Arab culture [5,8]. In Arab families, the father is the head of the household and holds a higher status than the mother [36]. Fathers may use their emotional bond with their children in order to control them. Indeed, the father’s authority is unquestioned in the family [50].

The role of Middle Eastern fathers is to ensure physical and economic needs are met [8]. Middle Eastern mothers typically have more responsibility around child rearing, discipline, education, and household duties. Therefore, when an adolescent misbehaves, the mother is blamed because of her lower status, and the adolescent’s behavior is seen as a reflection of the family’s reputation. Here, we found that mothers’ greater use of behavioral control was associated with lower levels of depressive symptoms and norm breaking. Mothers tend to structure their adolescents’ behavior. Fathers’ behavioral control is associated with fewer internalizing symptoms, as adolescents understand the rules and authority presented by their fathers, which helps them in predicting their fathers’ actions and expectations. Indeed, such predictions help adolescents to be less depressed.

We also expected that neighborhood support and assistance and adolescents’ gender may have unique effects on Palestinian refugee adolescents’ wellbeing. Past research shows that parents who live in high-risk neighborhoods tend to display less warmth, use harsh parenting styles, and discipline inconsistently compared to parents in safer neighborhoods (e.g., [51,52]). Rather, parenting had a stronger association with adolescents’ adjustment along the dimensions measured here.

One interpretation is that parents who are faced with chronic tension and stress or who do not have informal social cohesion with their neighbors lack support from their wider social context and thus engage in harsher and less-supportive parenting [53,54]. Alternatively, parents living in high-risk, low-income neighborhoods are prone to show less kindness and use harsher discipline [55], which may have a huge negative impact on adolescents. A second interpretation is that adolescents who live in an environment where lack of willingness from other residents (less social cohesion) to intervene, or lack of mutual trust and solidarity among neighbors, may increase the likelihood of parents’ harshness, which in turn increases the norm breaking and internalizing symptoms among adolescents [56]. Finally, it was surprising that adolescents’ perceptions of parental support from either mothers or fathers was not associated with any of the measures of adolescents’ adjustment.

## 5. Study Limitations and Future Directions

There are several limitations to the current study. First, we relied only on adolescents’ reports. Although adolescents’ experiences and perceptions of their parents are important, further research using multiple reporters would be very helpful in confirming the present results. Second, our data are cross-sectional and do not allow causal interpretations. Longitudinal research would be needed to determine the direction of effects. It is well known that families who are living in hazardous conditions tend to practice religion as a tool to mediate such internalizing symptoms. Therefore, it would be interesting to include religious activities and faith that impact their own perception of wellbeing. Moreover, families who participated in this study were all attending UNRWA schools, but there are other families who do not send their children to school, and they remain unstudied.

Despite some limitations, the present study makes a novel contribution to our understanding of parenting in refugee camps, as well as to the cross-cultural research on youth outcomes in the Middle East. Our results are consistent with a great deal of other research showing that living in poverty in physically challenging neighborhoods, which is typically the case for refugee families, is associated with poor parenting practices, such as psychological control and harsh discipline, and in turn with adolescents’ poor adjustment. Our study contributes to this literature by focusing on Palestinian youth in refugee camps and examining the role of fathers as compared to mothers. Palestinian Arab families are essentially rooted in having fathers who are responsible for all family members; moreover, adolescents must obey their fathers’ rules [57].

More studies of the distinct contributions of Arab mothers and fathers are needed, as very few studies of Arab families have focused on such comparisons. This research is needed to develop the most efficacious interventions for refugee parents with their adolescents, who are under significant stress. There are also other unanswered questions about how to incorporate culturally appropriate elements to parental stress interventions [58], and how and when to involve other family members such as siblings in such interventions [59]. It would be better to study the whole Arab family system, including cultural themes in such types of interaction, observed and grasped in a whole-systematic method. Overall, findings showed significant associations among socioeconomic and neighborhood physical conditions, parenting, and refugee adolescents’ adjustment, where dysfunctional parenting appears to impact the wellbeing and resiliency of children.

## Figures and Tables

**Table 1 ijerph-18-03649-t001:** Correlations among study variables.

	1.	2.	3.	4.	5.	6.	7.	8.	9.	10.	11.	12.
1. Adolescents’ Sex	1.00	0.03 **	0.03	−0.06	0.09	−0.05	0.14 **	−0.04	−0.07	0.03	0.06	−0.17 **
2. Adolescents’ Age	0.29 **	1.00	0.02	−0.04	0.13 *	−0.06	−0.09	−0.11	−0.12 *	0.12 *	0.03	−0.04
3. Parent Education	0.14 *	0.06	1.00	0.01	0.01	0.09	−0.06	−0.09	−0.09	−0.04	−0.3	0.03
4. Neighbor Assist	−0.06	−0.03	0.01	1.00	0.16 **	0.05	0.05	0.00	0.00	0.05	0.09	0.09
5. Physical Environ	0.08	0.13 *	−0.02	0.16 **	1.00	0.03	−0.04	0.44 **	0.14 **	0.07	0.16 **	0.12 *
6. Parent Support	0.09	0.00	0.19 **	0.08	−0.00	1.00	0.48 **	−0.38 **	−0.22 **	0.17 **	−0.17 **	−0.19 **
7. Behav Control	0.00	−0.11 *	0.14 **	0.05	0.05	0.42 **	1.00	−0.22 **	−0.08	0.29 **	−0.14 **	−0.24 **
8. Psych Control	−0.19 **	−0.09	−0.10	0.06	0.09	−0.40 **	−0.22 **	1.00	0.51 **	0.05	0.34 **	0.28 **
9. Harsh Parenting	−0.35 **	−0.17 **	−0.18 **	0.01	0.06	−0.35 **	−0.13 *	0.50 **	1.00	0.15 **	0.33 **	0.26 **
10. Self-Clarity	0.03	−0.12 *	−0.02	0.05	0.07	0.11 *	0.51 **	0.07	−0.08	1.00	0.26 **	0.07
11. Depression	0.06	0.03	−0.11 *	0.09	0.16 **	−0.18 **	−0.18 **	0.36 **	0.26 **	0.26 **	1.00	0.53 **
12. Norm Breaking	−0.18 **	−0.04	0.04	0.09	0.12 **	−0.26 **	−0.16 **	0.34 **	0.30 **	0.17	0.53 **	1.00

Notes: Assist = Assistance, Environ = Environmental, Beh = Behavioral Control, Psych = Psychological. Correlations above the diagonal = Mother perception and below the diagonal = Father perception. * *p* < 0.05, ** *p* < 0.01.

**Table 2 ijerph-18-03649-t002:** Hierarchical regression (final model) of parenting and neighborhood on adjustment.

	Internalizing Symptoms			Self-Clarity			Norm Breaking		
	β	ΔF	R^2^	β	ΔF	R^2^	β	ΔF	R^2^
Step 1—Demographics		1.26	0.01		2.15 ^+^	0.02		3.37 *	0.03
Adolescent Age	0.02			−0.09			0.00		
Sex (female)	0.12 *			0.03			−0.12 *		
Parents’ Education	−0.05			−0.00			0.08		
Step 2—Neighborhood		4.23 *	0.04		0.46	0.02		2.91 ^+^	0.04
Physical Condition	0.08			0.04			0.07		
Cohesion/Support	0.07			−0.01			0.05		
Step 3—Mom Parenting		10.86 **	0.16		6.84 **	0.11		9.29 **	0.13
Support	−0.01			0.05			−0.03		
Harsh Punishment	0.12 ^+^			0.05			0.07		
Psychological Control	0.10			0.04			0.10		
Behavioral Control	−0.02			0.30 **			−0.09		
Step 4—Dad Parenting		5.35 **	0.19		1.84	0.13		4.06 **	0.17
Support	−0.00			0.10			−0.12 ^+^		
Harsh Punishment	0.06			0.06			0.06		
Psychological Control	0.22 **			0.14 ^+^			0.12 ^+^		
Behavioral Control	−0.11 ^+^			−0.08			−0.06		

Note: Beta’s = at the final step. ^+^
*p* < 0.10, * *p* < 0.05, ** *p* < 0.01.

## Data Availability

Data and methods used in the research are presented in sufficient detail, where other researchers can replicate the work. We are ready to submit our Raw data publicly.

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
