# Peer review of "Palestinian Refugee Youth in Jordan: Parental Practices, Neighborhood Cohesion and Assistance, and Adolescent Wellbeing"

_ijerph, 2021, doi:10.3390/ijerph18073649_

Round 1
Reviewer 1 Report
Thank you for the opportunity to read this very interesting paper. There are some suggestions I would make regarding the paper prior to final consideration for publication.
For ease, I will cite the line or sections.
Line 32 - sentence finishing Arab peoples - a citation is needed here
lines 65-67 - sentence beginning Yes research in Arab countries - there appears to be a word missing - maybe in line 66 - during adolescence suggests harsh
line 83 - Arab family structure - for a reader not familiar with this, more detail about that family structure is needed. It might only be a sentence but we need to know what you are meaning there.
Line 88 - citation is needed
In the review of the literature, there is a confusion between refugee and refugee in camps. The refugee exists in a number of ways throughout the world. Thus, the literature being citied should refer to the specific population being researched. As well, the language needs to be constant such that the reader is clear you are referring to the specific population of Arab refugee families in refugee camps. This also needs to be very clear in the outline of the purpose of the present study.
There are also many models of refugee camps. It would be wise to give some description of the specific nature of the camp/community in this study. This might include some understanding of the social determinants of health for the population in this study.
Measures - can you help the reader to determine why these measures are valid to the population? Have they been normed on an Arab population? Have they been validated on a refugee population? I am concerned about these things as parenting is very cultural in its expression and thus, to what extent are these measures sensitive to the parenting styles described in the paper?
The norm breaking measure has specific concerns. Why are these norms not acceptable in Arab parenting? Why might they not be normative in the highly stressful environment of a refugee camp? I think the reader needs to know more about the context of this environment to know what adverse conditions exist that may lead to the behaviours measured and which might, in the context, be seen as adaptive or self protective.
Thank you for the explanation of the translation process.That was very helpful.
Line 118 - there should be a parenthesis mark after 1994 -there should not be a ; but rather , after Peteet
Ethics - There is no indication of an institutional ethics clearance. If that was not done, we will need to know why and then what steps
I hope these comments are helpful. The paper has strong potential for readers to better understand the challenges raised. Thank you for the work
Author Response
For ease, I will cite the line or sections.
- Line 32 - sentence finishing Arab peoples - a citation is needed here
We did so
- lines 65-67 - sentence beginning Yes research in Arab countries - there appears to be a word missing - maybe in line 66 - during adolescence suggests harsh
we did so
- line 83 - Arab family structure - for a reader not familiar with this, more detail about that family structure is needed. It might only be a sentence but we need to know what you are meaning there.
A statement has been added in line 92-94
- Line 88 - citation is needed
We added.
- In the review of the literature, there is a confusion between refugee and refugee in camps. The refugee exists in a number of ways throughout the world. Thus, the literature being citied should refer to the specific population being researched. As well, the language needs to be constant such that the reader is clear you are referring to the specific population of Arab refugee families in refugee camps. This also needs to be very clear in the outline of the purpose of the present study.
Thank you for your comments. We did take out all citations that are unrelated to this point. Besides, there are also many models of refugee camps. It would be wise to give some description of the specific nature of the camp/community in this study. This might include some understanding of the social determinants of health for the population in this study. In doing so, we added two statements (line 167 to 169) have been added describing the camps and its general structure.
- Measures - can you help the reader to determine why these measures are valid to the population? Have they been normed on an Arab population? Have they been validated on a refugee population? I am concerned about these things as parenting is very cultural in its expression and thus, to what extent are these measures sensitive to the parenting styles described in the paper?
Several people with different skills and disciplinary expertise participate in the translation, and, second, that a multistage process is used. questionnaire is newly designed for a cross-national comparative survey. Additionally, most of items were used are also adapted and existed in Arab countries.
- The norm breaking measure has specific concerns. Why are these norms not acceptable in Arab parenting? Why might they not be normative in the highly stressful environment of a refugee camp? I think the reader needs to know more about the context of this environment to know what adverse conditions exist that may lead to the behaviors measured and which might, in the context, be seen as adaptive or self protective.
Both inside and outside of refugee camps in Jordan, we do not consider hitting or biting to be norms. Besides, in the camps, there are schools and teachers who teach about cultural norms, morals, and norms. Of course, in the refugee camps their lives are different but there are police everywhere so it's not easy to break norms. So, if we are looking for any person who is breaking the norms, it should be understood as self-protective.
Thank you for the explanation of the translation process.
That was very helpful.
Line 118 - there should be a parenthesis mark after 1994 -there should not be a ; but rather , after Peteet
Thank you, we did so
Ethics - There is no indication of an institutional ethics clearance. If that was not done, we will need to know why and then what steps
I have added the following lines (181-183) for such matter. “Approval from the University of Jordan Human Subject’s Committee was acquired in order to collect data for the study. The researchers also obtained approval for research on human subjects from the UNRWA, also known in Jordan as the Office of Field in Jordan (OFJ).”
I hope these comments are helpful. The paper has strong potential for readers to better understand the challenges raised. Thank you for the work
Thank you so much for your comments.
-Best
Reviewer 2 Report
The manuscript needs improvement in aspects such as the following:
- Clarify the abstract to simplify the study and the main results.
-In the introduction, make sure the paragraphs are connected. The same ideas are presented repeatedly and there is no consistency. Add recent citations and recent research data. Quotations from the same author are abused and provide little diversity. Please add other inquiries.
-The authors lack justification of the variables that they will measure in the introduction.
- Missing information in the process.
-A section is missing that clarifies the statistical analyzes carried out.
-The procedure for selecting participants (inclusion and exclusion criteria) has not been clarified.
-The sample is so low, justify the reason.
-It is recommended to clarify the discussion section.
-A section with the conclusions of the manuscript is recommended
-It is recommended to take care of the structure of the template in the manuscript.
-Review the standard of journal citation.
Author Response
The manuscript needs improvement in aspects such as the following:
First: Clarify the abstract to simplify the study and the main results.
We did so and Thank you for your comments.
Second: In the introduction, make sure the paragraphs are connected. The same ideas are presented repeatedly and there is no consistency. Add recent citations and recent research data. Quotations from the same author are abused and provide little diversity. Please add other inquiries.
We did so
Third: The authors lack justification of the variables that they will measure in the introduction.
-our justification was through our own observation and the fact that after reading the article titled “Do Parenting and the Home Environment, Maternal Depression, Neighborhood, and Chronic Poverty Affect Child Behavioral Problems Differently in Different Racial-Ethnic Groups?”. We started thinking that we should study Palestinian refugees who are living in camps. Studying their behaviors after knowing the type of norm breaking that they are facing. Additionally, we have provided a clear justification for the particular variables in the paper.
Fourth: Missing information in the process.
-We have explained more in the method section.
Fifth: A section is missing that clarifies the statistical analyzes carried out.
We have mentioned the statically analyses in two paragraphs, We have wrote in line 241-244 as follows“ Because we were interested in the effects of different parenting dimensions on our three indices of adjustment, we elected not to use a latent variable approach and instead, included the individual parenting dimensions in hierarchical regression analyses conducted separately for the three adjustment variables.:
Besides, we have mentioned also at the beginning in line Means, SDs, and correlations among study variables are shown in Table 1.
Sixth: The procedure for selecting participants (inclusion and exclusion criteria) has not been clarified.
All students at the appropriate ages who attended schools in the camps were eligible for participation, and we included all students for whom we received parental consent. There were no exclusion criteria except lack of consent. We did not exclude students who had adjustment problems.
Seventh: The sample is so low, justify the reason.
We are surprised that you consider this a small sample. This is a good sized sample for studies of this kind, and especially so Palestinian youth in refugee camps.
Eight: It is recommended to clarify the discussion section.
We did so
Ninth: A section with the conclusions of the manuscript is recommended
We have discussed the conclusion in the result section. I did ask the editor as I already mentioned the discussion section where the result had been explained.
Tenth: It is recommended to take care of the structure of the template in the manuscript.
We did so
Eleventh: Review the standard of journal citation.
We did so

Round 2
Reviewer 1 Report
I have no further suggestions and recommend acceptance
Reviewer 2 Report
I consider that the article has improved in the current version.